# Preparation of β-Cyclodextrin Inclusion Complex and Its Application as an Intumescent Flame Retardant for Epoxy

**DOI:** 10.3390/polym11010071

**Published:** 2019-01-05

**Authors:** Xueying Shan, Kuanyu Jiang, Jinchun Li, Yan Song, Ji Han, Yuan Hu

**Affiliations:** 1School of Environmental and Safety Engineering, Changzhou University, Gehu Road 1, Changzhou 213164, China; xyshan@cczu.edu.cn (X.S.); 15189730617@163.com (J.H.); 2State Key Laboratory of Fire Science, University of Science and Technology of China, Jinzhai Road 96, Hefei 230026, China; 3School of Materials Science and Engineering, Changzhou University, Gehu Road 1, Changzhou 213164, China; 15195002735@163.com (K.J.); lijinchun88@163.com (J.L.); ysong@cczu.edu.cn (Y.S.)

**Keywords:** β-cyclodextrin, inclusion, flame retardant, epoxy

## Abstract

A new P-N containing the flame retardant, which was namely *N,N′*-dibutyl-phosphate diamide (DBPDA), was synthesized and it was assembled into the cavity of β-cyclodextrin (β-CD) to form an inclusion complex (IC). The structure and properties of IC were characterized by Fourier transform infraredspectroscopy (FTIR), wide-angle X-ray diffraction (WAXD), ^1^H nuclear magnetic resonance (^1^H NMR), scanning electron microscopy with X-ray microanalysis (SEM-EDS), differential scanning calorimeter (DSC) and thermal gravimetric analysis (TGA). ^1^H NMR and SEM-EDS were also used to identify the molar ratio of β-CD/DBPDA in IC and the results from the analyses indicated that their molar ratio was 1:1. In order to test the flame retardant effect of IC, it was added to epoxy (EP). IC was proposed to be able to act as an intumescent flame retardant (IFR) system in EP through a combination of β-CD and DBPDA properties during the combustion process. β-CD is a biomass carbon source, which has the advantages of environmental protection and low cost. Furthermore, DBPDA is both a source of acid and gas. When IC was heated, IC had the advantage of acting as both a carbon source and foam forming agent, while the DBPDA component were able to directly generate phosphoric acid and NH_3_ in situ. The impact of IC in low additive amounts on flame retardancy of EP was studied by the cone calorimeter test. When only 3 wt % IC was incorporated, the peak values of heat release rate (pHRR) and smoke production rate (pSPR) of EP were reduced by 22.9% and 33.3% respectively, which suggested that IC could suppress the heat and smoke release efficiently.

## 1. Introduction

Compared with the halogen-based flame retardants, intumescent flame retardant (IFR) systems have better flame retardant properties. The required additive amount of IFR is less and the amount of toxic corrosive gases generated in the combustion process is relatively smaller. In recent years, with the increasing requirements of flame retardants and people’s growing awareness of the need for environmental protection, IFR has drawn more and more attention in the field of flame retardants [1].

Generally, IFR systems mainly include three parts [2,3]: (1) Acid source: Inorganic acid or compounds that can generate acid species when they are heated to 100–250 °C, such as phosphoric acid and ammonium polyphosphate. (2) Carbon source: A type of polyhydroxy and carbon-containing compound, which can be dehydrated to form a char layer under the catalysis of an acid source, such as starch, cyclodextrin (CD) and pentaerythritol (3) Gas source: Some compounds that can generate volatile products under the conditions of heat or ignition, such as dicyandiamide, melamine and their derivatives. When IFR is heated in a polymer, the acid species generated from the acid source and carbon source undergo the process of esterification, which is catalyzed by the gas source. Water vapor generated from the esterification process and incombustible gases generated from the decomposition of the gas source cause the expansion of the system, which ultimately forms foam. Meanwhile, the carbon source and the esterification products cause charring. Finally, a porous char layer is formed. For polymer composites, this protective char layer is used as a physical barrier that can decelerate heat and mass transfer in the combustion process [4,5,6]. Therefore, IFR can achieve a good flame retardant effect. Unfortunately, the amount of IFR is usually larger if the better flame retardant effect is required.

As a bio-based carbon source, CD is obtained from the enzymatic degradation of starch and polysaccharides. CD is usually composed of 6–12 D-glucopyranose units. There are three main types of CD: α-CD (with 6 glucopyranose molecules), β-CD (with 7 glucopyranose molecules) and γ-CD (with 8 glucopyranose molecules). Among them, β-CD is the most widely used due to its low cost, appropriate molecular cavity size, good thermal stability and charring performance. In fact, β-CD has been used as a carbon source in combination with other materials to form IFRs that have been used as a flame retardant for polymers. For example, β-CD in combination with ammonium polyphosphate and melamine has been used as a flame retardant for polylactic acid [7]. β-CD, resorcinol bis(diphenyl phosphate) and phosphorus-containing polyacrylate were applied to improve the flame retardancy of latex film [8]. β-CD was used together with isopropylated triaryl phosphate ester to improve the flame resistance of polylactic acid/poly(methyl methacrylate) [9]. A thermoplastic polyester elastomer/β-CD/aluminum diethylphosphinate and melamine polyphosphate system was created and the flame retardant effect was determined in a previous study [10].

β-CD has a three-dimensional cavity that is hydrophilic in the external ring and hydrophobic in the internal ring. The chemical bond of β-CD has some flexibility at a certain extent and thus, it can act as a “host” molecule to accommodate different “guest”molecules in order to form a “host–guest” inclusion [11]. The “guest”molecules can be inorganic/organic compounds, organic/inorganic ions, benzene rings and non-polar hydrocarbon chains in the aliphatic compounds [12,13,14]. To date, β-CD inclusion complexes have a wide range of applications in medicine as the pharmaceutical carriers [15]. However, they have received less attention in the field of flame retardants. A β-CD/ferrocene inclusion complex was used in a polystyrene/IFR system [16]. An inclusion complex that was formed between β-CD and a commercial P-N containing flame retardant was processed by melting into polyethylene terephthalate films, which were tested for flammability [17]. β-CD/poly(propylene glycol)inclusion complex was chosen as a “green” carbon source in IFR and exhibited more effective carbonization and a higher degree of graphitic network than that of free β-CD [18]. Zhao et al. [19] prepared the inclusion complex of β-CD and *N,N′*-diamyl-p-phenylphosphonicdiamide. The results showed that 6 wt % inclusion could efficiently improve the thermal stability and flame retardancy of epoxy (EP). Zhang et al. [20] proved that the formation of an inclusion complex between β-CD and triphenyl phosphate could reduce the toxic effect of triphenyl phosphate while retaining its flame retardant properties. Triphenyl phosphate was released only during a fire when it was actually needed.

If the acid source and gas source are assembled in the cavity of β-CD to form an IFR system, it is expected that they will exhibit better flame retardant properties in polymers. In this work, a new halogen-free P-N containing flame retardant, which was namely *N,N′*-dibutyl-phosphate diamide (DBPDA), was synthesized. After this, it was assembled into the cavity of β-CD to form an inclusion complex (IC). IC was proposed to be an IFR system, in which β-CD was a biomass carbon source and DBPDA was both a source of acid and gas. When IC was heated, IC had the advantage of acting as both a carbon source and foam forming agent while the DBPDA component was able to directly generate phosphoric acid and NH_3_ in situ. The structure and properties of IC were characterized. Moreover, the impact of IC in alow additive amount (3 wt %) on flame retardancy of EP was studied by the cone calorimeter test.

## 2. Experimental

### 2.1. Materials

Phosphorus oxychloride (POCl_3_), pentaerythritol (PER), butylamine, β-CD, 4,4-diaminodiphenyl methane (DDM), trichloromethane, diethyl ether, acetonitrile and ethanol were purchased from Runyou Trading Co., Ltd., Changzhou, China. They were used without any further purification. The epoxy equivalent of EP was 0.44, which was obtained from Qingyi Chemical Materials Co., Ltd., Wuxi, China.

### 2.2. Synthesis of DBPDA

As for the intermediate product, the spirocyclic pentaerythritol bisphosphorate disphosphoryl chloride (SPDPC) was synthesized by the reaction of POCl_3_ with PER as previously reported [21]. The synthesis route of DBPDA is shown in Scheme 1a. In a three-neck flask, SPDPC was dispersed in acetonitrile at 0–5 °C. After this, butylamine was added dropwise and its molar amount was twice as much as the theoretical value. The whole mixture was reacted at 0–5 °C for 2 h before the reaction was continued at room temperature for 5 h. After the reaction finished, the product was filtered and washed three times with deionized water. After being dried in a vacuum oven, the white powder product was collected finally. The molecular length of DBPDA was about 2.1 nm according to the values of the bond length and bond angle.

### 2.3. Preparation of IC between β-CD and DBPDA

The preparation route of IC is described in Scheme 1b. β-CD was dissolved in deionized water at 65 °C to form the saturated aqueous solution. DBPDA was dispersed in acetone and slowly added dropwise into the prepared β-CD solution at 65 °C. The molar ratio of β-CD and DBPDA was 1:1. The mixture was stirred for 5 h before being left for 12 h at room temperature. This was then filtered, washed with deionized water and acetone before finally being dried in a vacuum oven to obtain IC. For comparison with IC, the physical mixture (PM) of β-CD and DBPDA was prepared as follows. β-CD and DBPDA (molar ratio 1:1) was dissolved in ethanol by stirring at room temperature. After this, the ethanol was evaporated.

### 2.4. Preparation of EP Sample and EP/IC Composite

In order to test the flame retardant effects of IC in EP, the EP/IC composite was prepared as follows. First, 3 wt % IC was dispersed into EP at 100 °C. The curing agent DDM was then dissolved into the EP/IC mixture. After DDM was totally dissolved, the final mixture was poured into the pre-heated polytetrafluoroethylene mold. The curing procedure was set as 80 °C/2 h and 160 °C/2 h. EP sample was prepared using the same method except for IC.

### 2.5. Characterizations

Fourier transform infrared (FTIR) spectra were measured with a Nicolet iS10 infrared spectrometer (Thermo Fisher Scientific Corporation, WA, USA). The KBr pellet pressing method was used to prepare the samples.

RigakuD/max2500 (Rigaku Corporation, Tokyo, Japan) was used to obtain wide-angle X-ray diffraction (WAXD) measurements. A Ni-filtered Cu Ka radiation source with λ = 1.544 Å was applied. The scanning range of 0–80°/2θ with a step size of 0.1° was chosen. The voltage and current were set to 45 kV and 40 mA, respectively.

^1^H nuclear magnetic resonance (^1^H NMR) measurements were obtained using a Avance III 400 M nuclear magnetic resonance spectrometer (Bruker Corporation, Karlsruhe, Germany). Dimethyl sulfoxide (DMSO) was used as the solvent.

Scanning electron microscopy with X-ray microanalysis (SEM-EDS) was conducted on SUPRA 55 (Zeiss Corporation, Jena, Germany).

The heat flow thermograms of β-CD, IC and PM under N_2_ atmosphere were tested by the differential scanning calorimeter (DSC, Q20, TA Instruments Corporation, New Castle, DE, USA). The settings were as follows: the temperature was 25–350 °C, and the heating rate was 10 °C/min.

Thermal gravimetric analysis (TGA) was performed on Q500 (TA Instruments Corporation). Samples were tested in the range of 25~700 °C under a N_2_ and air atmosphere at the heating rate of 10 °C/min.

According to ISO 5660-1:2002, the cone calorimeter tests were carried out with a cone calorimeter (FTT Corporation, East Grinstead, UK). The square specimen size was 100 mm × 100 mm × 3.2 mm.

## 3. Results and Discussion

### 3.1. Characterizations of IC

FTIR was used to show the interactions between β-CD and DBPDA. Figure 1a shows the FTIR spectra of DBPDA, IC and β-CD. The assignments of the main IR peaks of β-CD were: 3385 cm^−1^ (−OH), 2923 cm^−1^ (C−H of −CH_2_ and −CH), 1080, 1029 cm^−1^ (C−O−C), 1157 cm^−1^ (C−O−C) as well as 758 and 706 cm^−1^ (skeleton of β−CD). The assignments of the main IR peaks of DBPDA were: 1236 cm^−1^ (P=O), 1033 cm^−1^ (P-O-C), 1078 cm^−1^ (P-N), 2945 cm^−1^ (−CH_3_) as well as 3227 and 3428 cm^−1^ (–NH). Compared with the spectra of β-CD and DBPDA, IC showed greater similarity to the spectrum of β-CD rather than that of DBPDA. It indicated that DBPDA was encapsulated in the cavity of β-CD and the IR peaks of β-CD overlapped with most of the IR peaks of DBPDA. The frequency ranges of 1400~1100 cm^−1^ and 900~700 cm^−1^ are shown respectively in Figure 1a. In the range of 1400~1100 cm^−1^, the peak at 1240 cm^−1^ of IC did not appear in the spectrum of β-CD although this was found in the spectrum of DBPDA. In the range of 900~700 cm^−1^, the peak at 750 cm^−1^ of β-CD shifted to 760 cm^−1^ in the spectrum of IC. It indicated that DBPDA was successfully embedded into the cavity of β-CD.

A comparison between the WAXD spectra of β-CD and IC is shown in Figure 1b. Generally, the diffraction peaks of the host and guest molecules changed after the inclusion occurred. Three strong peaks at 2θ values of 6.3°, 11.5° and 17.2° were found in the β-CD spectrum. In comparison with the spectrum of β-CD, IC showed a completely different spectrum with three strong peaks at 2θ values of 10.7°, 12.5° and 19.2°. Because of the experimental re-crystallization process, the crystalline phase of β-CD changed, which showed the sharper and stronger diffraction peaks of IC. Compared with β-CD, there was a greater number of peaks in IC spectrum. The above phenomenon confirmed that a new complex formed during the inclusion process.

^1^H NMR was an effective tool to confirm the inclusion behavior and it could provide information about the stoichiometry of IC. ^1^H NMR spectra of DBPDA, β-CD and IC are given in Figure 2. Table 1 lists the σ and Δσ values of β-CD, IC and DBPDA. Σ represents the chemical shift values of protons. Δσ values were calculated as the differences between the σ of IC and σ of β-CD (or σ of DBPDA). For the β-CD spectrum, the σ values of H-3, H-5 and H-6 protons inside the β-CD cavity changed after DBPDA was embedded into the cavity as they shifted downfield by 0.011, 0.006 and 0.009 ppm, respectively. All σ values of H protons of DBPDA showed an upfield shift. Δσ values of β-CD and DBPDA indicated that DBPDA was embedded into the cavity of β-CD. It was noted that Δσ values of H’-1, H’-2, H’-3 and H’-4 on butyl chain of DBPDA were all 0.001 ppm. In addition, the integral area ratio of H-1 of β-CD (A)and H’-4 (B) was 3.4:2, indicating that one DBPDA molecule was included in one β-CD molecule, which meant that the molar ratio between β-CD and DBPDA was 1:1 in IC.

SEM-EDS was a useful tool to observe the morphology and detect the element types present on the sample surface. Figure 3 shows the results of SEM-EDS and elemental contents of IC. There was no phosphorus element in β-CD. However, phosphorus was detected in IC, which further indicated that the inclusion behavior between β-CD and DBPDA was successful. The average values of carbon, oxygen and phosphorus were 71.01 At%, 24.9 At% and 4.90 At% respectively. Using these results, we calculated that the molar ratio between β-CD and DBPDA was 1:1 in IC. SEM-EDS results were consistent with ^1^H NMR analyses.

Figure 4 shows the DSC curves of PM, IC and β-CD. There was a endothermic peak at 147 °C in the β-CD curve because of the phase change. No other peak appeared before 300 °C. As for PM, the phase change peak disappeared and the peak at around 277 °C was attributed to the degradation of PM. IC showed a completely different curve compared with β-CD and PM. Firstly, the phase change peak shifted to 132 °C and weakened apparently. Secondly, the degradation of IC occurred at 265 °C approximately, which was lower than that of PM. The reason was that the degradation of butylamine groups in DBPDA, which was able to accelerate the degradation of IC or PM. This effect of IC was stronger than PM. The above information showed that IC was not physically mixed during the inclusion procedure.

Figure 5 shows the TGA and DTG thermograms of β-CD, DBPDA, IC and PM under N_2_. According to the TGA data obtained from the samples, the initial degradation temperature (*T*_-5 wt %_), the temperature corresponding to maximumdegradationrate (*T*_max_) and char residue at 700 °C are listed in Table 2. *T*_-5 wt %_ and *T*_max_ of β-CD were about 289 °C and 328 °C, respectively. *T*_-5 wt %_ and *T*_max_ of DBPDA were 272 °C and 274 °C/302 °C. The degradation of butylamine groups occurred at 274 °C and the breakage of the DBPDA spiral ring structure happened at 302 °C. Compared to β-CD and DBPDA, *T*_-5 wt %_ of IC and PM was smaller due to the interactions between β-CD and DBPDA. For IC, this interaction was more apparent. The char residue of β-CD, DBPDA and PM at 700 °C was 8.4 wt %, 31.5 wt % and 23.1 wt %, respectively. It was noted that IC with 38.5 wt % residual char content had a greater amount of char residue compared to PM. The reason was that DBPDA as an acid source of the IFR system generated phosphoric substances in situ during heating, which could catalyze the char formation of β-CD. In addition, there was the esterification reaction between the degradation products of DBPDA and β-CD [22]. The above analyses indicated that the inclusion behavior enhanced the catalysis and esterification effect between DBPDA and β-CD, resulting in better charring ability of IC.

### 3.2. Impact of IC on Thermal Stability of EP

The thermal degradation curves with the one-stage formation of EP and EP/IC (3 wt %) composite under N_2_ are shown in Figure 6. With the addition of 3 wt % IC, *T*_-5 wt %_ of EPdecreased from 359 °C to 338 °C. This suggested that IC had a significant impact on the initial thermal degradation of EP as IC had a lower thermal degradation temperature (*T*_-5 wt %_: 251 °C). *T*’_max_ of EP/IC almost did not change in comparison with EP (*T*_max_ for EP was 387 °C and *T*’_max_ for EP/IC was 389 °C). The amount of char residue at 700 °C only increased slightly when 3 wt % IC was added (17.1 wt % for EP/IC and 15.3 wt % for EP).

Figure 7 shows the thermal-oxidative behavior of EP and EP/IC composite under air. EP had a two-stage TGA curve and *T*_-5 wt %_ was 344 °C. EP/IC had a three-stage TGA curve and *T*’_-5 wt %_ was 334 °C, which was lower than that of EP. This result was induced by the thermal-oxidative behavior of IC. *T*_max1_ and *T*_max2_ of EP were 375 °C and 560 °C, respectively, while *T*’_max1_, *T*’_max2_ and *T*’_max3_ of EP/IC were 367 °C, 397 °C and 588 °C respectively. The decomposition products of DBPDA in IC had phosphoric acid compounds, which could catalyze the decomposition of EP at the first step. However, IC additive improved the thermal stability of EP at the second step. There was almost no char residue at 700 °C for two samples as it was only 1.3 wt % for EP and 2.5 wt % for EP/IC.

### 3.3. Impact of IC on Flame Retardancy of EP

The cone calorimeter test was an effective way to evaluate the combustion behavior of materials, and this provided some characteristic data relevant to real fire. Figure 8 shows the heat release rate (HRR) vs. time, the total heat release (THR) vs. time, smoke production rate (SPR) vs. time and the total smoke production (TSP) vs. time curves of EP and EP/IC. pHRR and pSPR represent the peak values of HRR and SPR curves, respectively. The corresponding data were shown in Table 3. Compared with EP, pHRR and pSPR data of EP/IC decreased from 980 KW/m^2^ and 0.6 m^2^/s to 756 KW/m^2^ and 0.4 m^2^/s with the reduction of 22.9% and 33.3%, respectively. THR and TSP data of EP/IC also decreased from 81 MJ/m^2^ and 14.8 m^2^ to 75 MJ/m^2^ and 13.4 m^2^ with the reduction of 7.4% and 9.5%, respectively. These results suggested that the combustion decreased obviously with a low additive amount of IC (3 wt %) in EP. In other words, IC could improve the flame retardancy of EP efficiently, which ultimately decreases heat release and smoke production.

Figure 9 shows the digital photos of char residue after cone calorimeter tests. For EP, only a small amount of char residue was left. In comparison with EP, there was a greater amount of char residue. Actually, IC acted as an IFR in EP. DBPDA was both an acid and gas source, which was able to accelerate the char formation. β-CD was a carbon source because of its polyol structure. When EP/IC burned, IC could help EP to form a large number of high-quality char layers. This char layer acted as a physical barrier and provided the condensed phase flame-retardant, further improving the flame retardancy of EP. Although the addition of IC in EP was only 3 wt %, the flame retardant effect is obvious after looking at Figure 8 and Figure 9.

## 4. Conclusions

In this work, a new P-N containing flame retardant DBPDA was synthesized and it was assembled into the cavity of β-CD to form an inclusion complex IC. The results of the characterizations revealed that an IC was prepared successfully. The inclusion ratio of β-CD and DBPDA was 1:1 according to the results of ^1^H NMR and SEM-EDS. TGA under N_2_ showed that the charring ability of IC was higher than that of PM as the char residue content of IC was 38.5 wt % at 700 °C while that of PM was 23.1 wt %. The initial thermal degradation temperature of IC was lower due to the strong interaction between β-CD and DBPDA.

To investigate the flame retardant effect of IC, 3 wt % additive amount of IC was incorporated into EP. TGA showed that IC mainly affected the initial degradation or decomposition temperature of EP. Compared with EP, the cone calorimeter tests showed pHRR and pSPR were reduced by 22.9% and 33.3%, respectively. THR and TSP also reduced by 7.4% and 9.5%, respectively. The results indicated that a low additive amount of IC had a good flame retardant effect. Combined with the digital photos of char residue, IC could help EP to form a large number of high-quality char layers. This char layer acted as a physical barrier and provided the condensed phase flame-retardant, further improving the flame retardancy of EP.

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
