# Peer review of "Preparation of β-Cyclodextrin Inclusion Complex and Its Application as an Intumescent Flame Retardant for Epoxy"

_polymers, 2019, doi:10.3390/polym11010071_

Round 1
Reviewer 1 Report
The necessity of doing this work should be obviously dealt with in both ABSTRACT and NTRODUCTION. Therefore, Introduction and abstract need revision in order to show clearly and highlight the necessity of research. What’s the main objective and interest of “host” CD? What’s the effect of FR on curing behavior/kinetics epoxy composites ? Cone calorimetry data should be summarized in a table. The difference between pristine EP and EP/IC sample in cone calorimetry is not significative. LOI and/or UL tests could be useful. The digital photos are not really informative. Authors could make more analysis on char residue.
Author Response
Dear Reviewers,
Thank you for your comments.
The response file is in the attachment.
Best regards.
Xueying Shan

Reviewer 2 Report
An inclusion complex in beta cyclodextrin of a novel phosphorated P-N containing fire retardant has been synthetized and characterized. Then, it has been used as a fire retardant in epoxy resin.
My opinion is that this paper can be published, provided that a major revision is carried out.
One of the concern is about the real formation of an inclusion complex. One of the best proof of IC formation comes from the disappearance of the melting point of the guest molecules in the DCS of the IC. Here DSC of pure guest molecule is not reported and so this relevant comparison cannot be done.
The protonic NMR shift factors for DBPDA in IC and in the neat products are so small as small also are those for BCD (At least very much smaller in comparison to the shift reported in literature for other IC with BCD). So, interaction BCD/DBDPA should be quite low.
However, I wonder what the advantage in using IC BCD-DBDPA instead of their physical mixture could be as far as flame retandant behavior is concerned in this particular case. In effect the charred residue (condensed phase effect) do not appear to increase (see TGA, fig. 6), but it will be interesting to report the gravimetric data for come calorimeter also which are missed. The gas phase active molecule is released at lower temperature than those at which resin degradation occurs, even lower than that of the physical mixture.
Cone calorimeter experiment confirms some fire retardant action however better behavior will certainly be obtained by increasing the amount of IC additive. Usually a good intumescent char should be physically resistant and thermally insulant, however, in Fig 9 the residual char appear split and broken so leaving the flammable volatile escaping into the flame.
In addition:
It will be better to add some detail in the experimental part, for example the formula of EP and the ratio between EP and DDM, which controls the crosslinking density of the resin.
Please correct BCD formula in Figure 2 (H instead of OH at C1)
I’m not sure that peak at 147 °C in DSC of Figure 4 can be attributed to dehydration reaction in BCD (No weight loss occurs in this range of temperature (TGA), melting or phase change could be instead.
I do not agree with some IR attribution mainly 1157 cm-1 (should be C-O-C and not C-C)
Author Response

(The authors gave the same response as above.)

Round 2
Reviewer 1 Report
All comments are considered by authors.
Reviewer 2 Report
I think that this paper can be now published as it is. English revision is needed